# Oral Vaccine Formulation for Immunocastration Using a Live-Attenuated *Salmonella* ΔSPI2 Strain as an Antigenic Vector

**DOI:** 10.3390/vaccines12121400

**Published:** 2024-12-12

**Authors:** Sergio A. Bucarey, Lucy D. Maldonado, Francisco Duarte, Alejandro A. Hidalgo, Leonardo Sáenz

**Affiliations:** 1Centro Biotecnológico Veterinario, Biovetec, Departamento de Ciencias Biológicas Animales, Facultad de Ciencias Veterinarias y Pecuarias, Universidad de Chile, Santa Rosa 11735, La Pintana, Santiago 8820808, Chile; lucymaldonadob@gmail.com (L.D.M.);; 2Escuela de Química y Farmacia, Facultad de Medicina, Universidad Andres Bello, Sazié 2320, Santiago 8370134, Chile; 3Laboratorio de Vacunas Veterinarias, Departamento de Ciencias Biológicas Animales, Facultad de Ciencias Veterinarias y Pecuarias, Universidad de Chile, Santa Rosa 11735, La Pintana, Santiago 8820808, Chile; lsaenz@uchile.cl

**Keywords:** oral immunization, GnRH, *S.* Typhimurium, ΔSPI2

## Abstract

Immunization against Gonadotropin-Releasing Hormone (GnRH) has been successfully explored and developed for the parenteral inoculation of animals, aimed at controlling fertility, reducing male aggressiveness, and preventing boar taint. Although effective, these vaccines may cause adverse reactions at the injection site, including immunosuppression and inflammation, as well as the involvement of laborious and time-consuming procedures. Oral vaccines represent an advancement in antigen delivery technology in the vaccine industry. In this study, a *Salmonella enterica* serovar Typhimurium (*S.* Typhimurium) mutant lacking the pathogenicity island 2 (*S.* Typhimurium ΔSPI2) was used as a vehicle and mucosal adjuvant to deliver two genetic constructs in an attempt to develop an oral immunological preparation against gonadotropin hormone-releasing hormone (GnRH). *S.* Typhimurium ΔSPI2 was transformed to carry two plasmids containing a modified GnRH gene repeated in tandem (GnRXG/Q), one under eukaryotic expression control (*pDNA*::GnRXG/Q) and another under prokaryotic expression control (pJexpress::GnRXG/Q). A group of three male BALB/c mice were orally immunized and vaccination-boosted 30 days later. The oral administration of *S.* Typhimurium ΔSPI2 transformed with both plasmids was effective in producing antibodies against GnRXG/Q, leading to a decrease in serum testosterone levels and testicular tissue atrophy, evidenced by a reduction in the transverse tubular diameter of the seminiferous tubules and a decrease in the number of layers of the seminiferous epithelium in the testes of the inoculated mice. These results suggest that *S.* Typhimurium ΔSPI2 can be used as a safe and simple system to produce an oral formulation against GnRH and that Salmonella-mediated oral antigen delivery is a novel, yet effective, alternative to induce an immune response against GnRH in a murine model, warranting further research in other animal species.

## 1. Introduction

Surgical castration has been a frequently used method in veterinary medicine to address problems associated with sexual maturity in animals. It has been employed to control animal overpopulation, reduce male aggressiveness, prevent sexual odor in pork, and improve body growth rates in production animals, among other uses. However, this practice presents several drawbacks, mainly related to animal welfare, such as the pain it causes to the castrated individual, particularly in intensive production systems where anesthesia is usually not applied, in addition to possible infections due to poor postoperative management and even the death of the animal.

Gonadotropin-Releasing Hormone (GnRH) is a decapeptide that plays a central role in mammalian reproduction [1,2,3]. It is secreted from the mediobasal portion of the hypothalamus and transported via axons to the median eminence, where it is released into the hypothalamic-pituitary portal circulation, inducing the release of luteinizing hormone (LH) and follicle-stimulating hormone (FSH) from the anterior pituitary [4], which promote ovarian follicle maturation in females and spermatogenesis in males [1].

For many years, efforts have been made to immunoneutralize GnRH, blocking its entry into the pituitary [5]. This technique, known as immunocastration, allows for blocking the hormone’s action through an immunological preparation, eliminating the production and release of the sex steroids responsible for reproductive capacity and sexual behavior [6]. The mechanism of action involves preventing the interaction of GnRH with receptors on gonadotropic cells located in the pituitary by binding the hormone to specific antibodies, thereby inhibiting the entire subsequent hormonal cascade [7,8].

GnRH is an ideal candidate for inducing immunocastration in animals because a single product is effective in both males and females, and its amino acid sequence is identical across all mammals [7,9]. However, the main challenge in inducing an immune response against this hormone lies in its low immunogenicity due to its small size, which is insufficient to act as an antigen [1].

It is for this reason that, to effectively induce the production of neutralizing antibodies against GnRH, it must be coupled to carrier molecules [10,11]. To generate an effective immune response, it has been conjugated with highly immunogenic molecules, such as bovine serum albumin, ovalbumin, and keyhole limpet hemocyanin (KLH). Additionally, some formulations include pathogen-associated molecular patterns (PAMPs), which stimulate antigen-presenting cells to generate an adaptive immune response of effective magnitude and duration [1,12].

Immunocastration has several implications in veterinary medicine. Primarily, it is used to replace surgical castration, a practice associated with pain and health risks in animals, especially in production settings where anesthesia is commonly not applied. This affects animal welfare and can lead to inflammation, bleeding, hernia predisposition, and significant economic losses [13]. In contrast, immunocastration is considered an ethical, practical, cost-effective, and efficient procedure [14,15]. In the swine industry, immunocastration is used to eliminate sexual odor, a sensory defect affecting meat from intact males caused by two compounds: androstenone and skatole, which accumulate in male pig adipose tissue upon reaching puberty, imparting an unpleasant odor and flavor to the meat [16,17]. In cattle raised for meat production, immunization against GnRH has been proposed as a late-stage strategic castration method for males, improving animal welfare while providing the benefits of surgical castration. It serves as a key management tool to enhance carcass quality through increased fat deposition and reduce aggressive and sexual behavior [18,19]. In other species, such as dogs and cats, immunocastration can be applied mainly as a method for controlling reproduction and reducing male aggression [4,20]. In addition, control of aggressive behavior has also been seen in Holstein bulls [21].

Currently, there are nine immunocastration vaccines on the market. Among these, *Improvac*, also marketed as *Innosure* and produced by Zoetis for use in pigs, stands out with an efficacy duration of eight to twelve weeks. This vaccine consists of a synthetic GnRH analog with an aqueous adjuvant [22]. Another key product is *Bopriva*, also from Zoetis, intended for cattle, containing a modified GnRH peptide in an aqueous adjuvant specific to livestock, with an efficacy duration of approximately fifteen weeks [4]. All vaccines must be administered subcutaneously, which presents handling challenges in intensive production systems where numerous animals must be inoculated in a short time while ensuring operator safety and avoiding accidental self-inoculation. Therefore, there is a need for an alternative that facilitates administration, such as the formulation of an oral vaccine.

Oral vaccines offer several advantages over conventional injectable vaccines: they are easy to administer, safe to handle (since they do not require needles), suitable for mass immunization, stable without refrigeration (if lyophilized), and have relatively low production costs [23,24]. In this way, oral vaccines meet the criteria for an ideal vaccine, which include no use of syringes, no need for a cold chain for distribution, and the application of a single dose to exert its effect. Vaccines using attenuated live strains of *Salmonella* as vectors meet these requirements [25].

Oral administration of *Salmonella* results in colonization of Peyer’s patches through M cells in the intestinal tract of mammals and colonization of mesenteric lymph nodes, liver, and spleen, resulting in both local and systemic humoral and cellular immune responses [26,27]. To interact with the host, *Salmonella* has developed a strategic mechanism involving specialized macromolecular organelles called type III secretion systems (TTSS), whose central function is the translocation of bacterial proteins into eukaryotic cells. This system mediates the transfer of bacterial effector proteins capable of modulating cellular functions in the host cell [28]. *Salmonella enterica* encodes two TTSS, located on different pathogenicity islands: one is found in pathogenicity island 1 (SPI1) and is responsible for the initial interaction of the bacteria with intestinal epithelial cells, inducing actin cytoskeleton reorganization and internalization of *S. enterica*; the other TTSS is located in pathogenicity island 2 (SPI2) and allows the intracellular survival and replication of the bacteria required to produce a successful systemic infection [28]. For these bacteria to be used as oral vaccine vectors, it is necessary to significantly reduce their virulence through various attenuation methods [29,30]. Thus, mutations in SPI2 result in significant attenuation of *Salmonella* virulence and a defect in its ability to cause systemic infection due to its inability to survive within macrophages [28].

Attenuated strains of *Salmonella* expressing heterologous antigens offer a valuable alternative to conventional vaccines, as they induce effective adaptive humoral, cellular, and mucosal immune responses, as demonstrated in various animal models, including rats, mice, rabbits, and pigs [23,25,31] and even in human volunteers [32]. Moreover, the existence of the Ty21a vaccine, developed with live attenuated *Salmonella enterica* serovar Typhi against typhoid fever, which has been approved by the United States FDA for human use, serves as an important safety benchmark for future research in this area [33]. Furthermore, *Salmonella* has been shown to be useful for delivering DNA vaccines, where the attenuated bacteria carry a plasmid encoding eukaryotic antigens, which, although not expressed in the bacteria, can be delivered to the host macrophage, resulting in the expression and presentation of the antigen by the host’s immune system very efficiently [31,34].

In this work, the concept of oral immunization against GnRH was tested in a murine model. An attenuated *S. enterica* serovar Typhimurium strain lacking SPI2 (*S.* Typhimurium ΔSPI2) carrying the recombinant *GnRXG/Q* antigen was used as a vector under two different formulations: one as a constitutive producer of the recombinant antigen and the other as a carrier of the *GnRXG/Q* gene under a eukaryotic replication system. The proof of concept for these two formulations was conducted by evaluating the adaptive immune response generated in male mice by measuring antibodies against the antigen, testosterone levels, and testicular changes after oral administration of two doses of the formulation. Although still largely in the experimental stage, the use of oral vaccines for immunocastration could represent a breakthrough in animal welfare, providing a humane, cost-effective, and efficient alternative to surgical castration and injectable immunocastration. This study provided relevant data on adaptive immunity against GnRH under an oral immunization scheme. Moreover, the efficacy and safety of this proof of concept allow for the projection of these results toward future preclinical trials.

## 2. Materials and Methods

The study was conducted at the Veterinary Biotechnology Center, Biovetec, Faculty of Veterinary and Animal Sciences, University of Chile (Santiago, Chile).

### 2.1. Transformation of Attenuated S. Typhimurium with Plasmids Carrying the GnRXG/Q Gene

To transform attenuated *S.* Typhimurium, two plasmids containing a modified tandem-repeated GnRH gene were used: the *GnRXG/Q* antigen [5]. For transformation, an isolated colony of *S.* Typhimurium 14028s ΔSPI2 (obtained by the deletion of the full length of the SPI2, by λ Red recombination system [35]) was cultured in LB-broth (10 g tryptone, 5 g yeast extract, and 5 g sodium chloride in 1 L of distilled water) with vigorous shaking for 18 h at 37 °C. The bacteria were made transformation-competent by repeating 5 cycles of centrifugating at 10,000× *g* for 5 min and resuspending the bacterial pellet with 10 mL of sterile water. After a final centrifugation, the bacterial sediment was resuspended in 200 µL of sterile water, and the bacteria transformed by electroporation at 2.2 kV, 25 µF, and 200–400 Ω with each plasmid: *pJexpress::GnRXG/Q* (DNA TwoPointO, Inc., Newark, CA, USA) for prokaryotic expression, hereafter referred to as *pJEX*, and *GnRX GQ_OptMamm_V3::GnRXG/Q* (DNA 2.0, CA, USA) for eukaryotic expression, hereafter referred to as *pDNAX3*. Positive transformation was selected on LB agar with 50 µg/mL kanamycin for the *S.* Typhimurium ΔSPI2/pJEX strain and LB agar with 100 µg/mL ampicillin for the *S.* Typhimurium ΔSPI2/pDNAX3 strain.

### 2.2. Intracellular Survival Assay of S. Typhimurium in Murine Macrophages RAW264.7

An intracellular survival assay in macrophages was performed to verify vector attenuation. The assay was based on the one described by [36]. RAW264.7 cells (provided by Dr. Gonzalo Cabrera, Faculty of Medicine, University of Chile) were cultured at 37 °C and 5% CO_2_ in RPMI medium enriched with 10% fetal bovine serum, 1% L-glutamine, 1% sodium pyruvate, and 1% streptomycin and penicillin. The cells were transferred to 24-well culture plates until reaching 80% confluence. For the assay, the wild-type *S.* Typhimurium 14028s strain (from Dr. Stanley Maloy’s strain collection, University of San Diego, San Diego, CA, USA) was used as the control, and the mutant strain *S.* TyphimuriumΔSPI2/pJEX was grown in LB-broth under anaerobic conditions at 37 °C until an OD600 of 0.2 units was reached. A total of 200 µL of this culture was centrifuged at 13,000× *g* for 10 min, and the supernatant was discarded. The bacterial sediment was resuspended in 1 mL of enriched RPMI. Then, 50 µL of the bacterial suspension was added to each well, aiming for a multiplicity of infection of 50 bacteria per cell. Nine wells with cell monolayers were used for each bacterial strain. The culture plate was incubated in a 5% CO_2_ atmosphere at 37 °C for 1 h to allow bacterial entry into the cells (invasion). Immediately afterward, the monolayers were washed twice with sterile PBS, and 100 µL of enriched RPMI supplemented with 200 µg/mL gentamicin was added to each well. The plate was incubated for 2 h to eliminate extracellular bacteria. The medium was removed, and the monolayers were washed twice with sterile PBS. Three wells were treated with 0.5% sodium deoxycholate in sterile PBS to lyse the cells. The number of intracellular bacteria (CFU at t_0_) was determined by serial dilution and seeding of the cell lysate on LB agar plates. For the remaining six wells of each bacterial strain, fresh RPMI medium with 25 µg/mL gentamicin was added, and the plates were incubated for 24 and 48 h to allow intracellular proliferation. The monolayers were lysed, and the recovered bacteria (CFU at t_24–48 h_) were counted. The survival percentage of each strain was calculated using the following equation:
$$\%\;of\;survival\;at\;time\;n=\frac{CFU\;recovered\;at\;t_{n}}{CFU\;inoculated\;at\;t_{0}}\times 100$$

### 2.3. Antigen Expression and Detection via Western Blot Analysis

To demonstrate the expression of the *GnRXG/Q* antigen in the recombinant strain *S.* Typhimurium ΔSPI2/pJEX (prokaryotic expression plasmid), a Western blot analysis was performed using the purified *GnRXG/Q* antigen as a positive control. An isolated colony of the recombinant strain *S.* Typhimurium ΔSPI2/pJEX was cultured in 10 mL of LB broth and incubated with vigorous shaking for 18 h at 37 °C. The bacteria were then reinoculated into 90 mL of LB broth and incubated at 37 °C until an OD_600_ > 0.6 was reached. Once the optical density was achieved, 1 mM IPTG was added to induce protein synthesis, and the culture was incubated at 37 °C with shaking for 18 h. The cells were harvested by centrifugation at 10,000× *g* for 15 min at 20 °C, resuspended in binding buffer (20 mM sodium phosphate, 0.5 M NaCl_2_, 40 mM imidazole, pH 7.4) (4 mL per gram of pellet), and lysed by sonication for 8 cycles of 15 s with 15-s intervals at an amplitude of 40%. The cells were then centrifuged at 10,000× *g* for 10 min at 20 °C and resuspended in lysis buffer (50 mM Tris-HCl, 10% glycerol, 10 mM EDTA) with urea (2 mL per gram of pellet).

The bacterial whole cell extracts, equivalent to 30 µg of protein, were subjected to SDS-PAGE electrophoresis on 10% acrylamide gels at 150 V for 1 h. Subsequently, the gel was wet transferred to a nitrocellulose membrane for 30 min at 250 mA. After the transfer, the membranes were blocked with 5% skim milk in 1× PBS for 18 h at room temperature. The membranes were incubated with primary antibody against *GnRXG/Q* antigen before washing twice with TBST (1X TBS, 0.05% Tween 20) and incubating with anti-mouse IgG antibody conjugated to peroxidase enzyme (Jackson Immunoresearch™, West Grove, PA, USA) diluted in ELISA buffer (1X PBS, 3% BSA). The membrane was washed five times with TBST and developed with the development buffer (5 mL phosphatase buffer, 33 µL BCIP, 16.5 µL NBT). Once the color developed, the membrane was washed with distilled water, and the band was photographed under a transilluminator.

### 2.4. Preparation of Inoculum for Immunization

The bacterial strains *S.* Typhimurium 14028s ΔSPI2/pJEX and *S.* Typhimurium 14028s ΔSPI2/pDNAX3 were cultured in 10 mL of LB broth with 50 µg/mL kanamycin and 10 mL of LB broth with 100 µg/mL ampicillin, respectively, for 18 h at 37 °C. They were then reinoculated into 90 mL of LB broth with 50 µg/mL kanamycin and 90 mL of LB broth with 100 µg/mL ampicillin, respectively, and incubated at 37 °C to an OD_600_ of 0.6–1.0. The *S.* Typhimurium 14028s ΔSPI2/pJEX culture was induced with 1 mM IPTG at 37 °C with shaking for 18 h. The cultures were serially diluted in LB broth to reach a concentration of 10^9^ CFU per inoculum. They were centrifuged for 10 min at 10,000× *g*, and the bacterial pellet was resuspended in 4 parts PBS and 1 part of 7.5% sodium bicarbonate at pH 8.5 in a total volume of 200 µL, before inoculating mice as described by [37].

### 2.5. Mouse Immunization

Twelve male BALB/c mice (6–8 weeks old) were housed in the animal maintenance unit at Biovetec laboratory and fed ad libitum with food and water in a controlled environment (23 ± 1 °C, indirect light 12 h on/12 h off). The animals were housed in specially designed polycarbonate cages equipped with stainless steel tops containing feeders and water bottles. Care was taken to ensure adequate space and ventilation for each animal. The feeding, hydration, and cleaning of the mice were supervised daily as part of this research project.

The animals were randomly divided into four experimental groups, each consisting of three mice. Group A was orally inoculated with 10^9^ CFU of the live attenuated bacterial strain *S.* Typhimurium 14028s ΔSPI2/pJEX, which expressed the *GnRXG/Q* antigen. Group B was orally inoculated with 10^9^ CFU of the live attenuated bacterial strain *S.* Typhimurium 14028s ΔSPI2/pDNAX3, which carried the *GnRXG/Q* gene. Group C served as the positive control and was subcutaneously inoculated with 100 µg of affinity-purified *GnRXG/Q* protein plus aluminum as an adjuvant. Group D received 200 µL of PBS orally and served as the negative control. The mice were fasted for 4 h before immunization. The inoculum was administered orally using a plastic pipette. A booster was administered to all groups 30 days after the first immunization.

Blood samples (200 µL) were collected from the external saphenous vein on days 0, 15, 30, 45, and 60 post inoculation. Blood samples were collected, allowed to coagulate at room temperature, and stored at 4 °C overnight before being centrifuged at 2500× *g* for 20 min at 4 °C. The resulting sera were collected and stored at −20 °C until use.

The clinical trials with mice were monitored to evaluate any potential side effects that could compromise the welfare of the experimental animals. The initial reference for clinical trial monitoring was based on the guidelines for choosing endpoint criteria for experimental animals as described by the Canadian Council on Animal Care [38].

### 2.6. Measurement of Antibody Levels Against GnRXG/Q

The IgG anti-*GnRXG/Q* antibody titers were determined by ELISA using the following protocol: ELISA plates (96 wells) were coated with the *GnRXG/Q* antigen diluted in Coating Buffer (0.15 M Na_2_CO_3_–0.35 M NaHCO_3_, pH 9.6). A total of 50 µL of the diluted antigen was added to each well, except for the controls. The plates were incubated for 18 h at 4 °C without shaking. The solution was then removed, and the wells were washed twice with 200 µL of washing buffer (1X PBS and 0.02% Triton 100X). To block nonspecific binding sites, 200 µL of blocking buffer (5% skimmed milk diluted in washing buffer) was added to each well. The plates were incubated for 18 h at 4 °C without shaking. The solution was then removed, and the plates were washed twice with 200 µL of washing buffer per well. For the antigen-antibody reaction, 100 µL of mouse serum diluted 1:250 in ELISA diluent buffer (0.5% skimmed milk plus washing buffer) was added to each well, and the plates were incubated for 2 h at 37 °C without shaking. The plates were then washed five times with washing buffer. To detect the antigen-antibody complex, 100 µL of anti-mouse IgG antibody conjugated to peroxidase (Jackson Immunoresearch, West Grove, PA, USA) diluted 1:5000 in ELISA diluent buffer was added to each well. The plates were incubated for 45 min at 37 °C without shaking. Finally, the plates were washed five times with 200 µL of washing buffer per well. The reaction was detected by adding 100 µL of tetramethylbenzidine TMB-ELISA substrate solution to each well, and the plates were incubated for an additional 10 min at 37 °C without shaking. The reaction was stopped with 100 µL of stop solution (1.5 M H_2_SO_4_ in deionized water at room temperature). The absorbance was read at 450 nm in an ELISA microplate reader (Bio-Rad Laboratories, Inc., Hercules, CA, USA). All samples were analyzed in duplicate. The results were expressed as mean ± standard deviation (SD) of absorbances measured at 450 nm.

### 2.7. Measurement of Serum Testosterone Levels

The quantitative determination of testosterone in the serum samples obtained from the study animals was performed using the commercial ELISA kit for testosterone (IBL-America, Inc., Minneapolis, MN, USA). The reagents included in the kit were allowed to reach room temperature before starting the assay. The washing solution was prepared with deionized water according to the manufacturer’s instructions.

A total of 25 µL of each sample was dispensed into each well, in parallel with the standard and control reagents provided with the kit. Then, 200 µL of the enzyme-conjugated reagent was added to each well and mixed for 10 s. The plate was incubated for 60 min at room temperature, after which the wells were emptied, and the unbound conjugate was removed by washing the plates three times with 400 µL of washing buffer per well.

For the development reaction, 200 µL of substrate solution was added to each well, and the plate was incubated for 15 min at room temperature. The enzymatic reaction was stopped by adding 100 µL of stop solution to each well. The absorbance was read at 450 nm in an ELISA microplate reader (Bio-Rad Laboratories, Inc.).

### 2.8. Histological Evaluation of Testes

The mice were euthanized by CO_2_ exposure after the last blood samples were collected. Testicles were extracted and fixed in 10% formalin. In the facilities of the Pathology Department of the Faculty of Veterinary and Animal Sciences, University of Chile, the testes were sectioned at 5 µm thickness, stained with hematoxylin-eosin, and analyzed by optical microscopy to observe the transverse tubular diameter of the seminiferous tubules and the number of cell layers. The transverse tubular diameter corresponds to the average of two measurements taken diametrically opposed.

### 2.9. Statistical Analysis

To analyze the data obtained, one- and two-factor analysis of variance (ANOVA) was performed using GraphPad Prism software (version 9.3). The sample means of antibody and testosterone levels between the two vaccine formulations under study and the positive and negative control groups were compared, as well as the transverse tubular diameters of the seminiferous tubules and the number of seminiferous epithelial layers in both testicles of the mice. Immunization was considered effective if there were statistically significant differences with the control groups. The statistical analyses of the survival assays were similarly performed using GraphPad Prism software^®^, with Student’s *t*-test used to analyze significant differences between the mutant strains and the wild-type strain.

## 3. Result

### 3.1. Intracellular Survival of S. Typhimurium in Murine Macrophages RAW264.7

An in vitro assay was conducted to evaluate the survival of *S.* Typhimurium 14028s ∆SPI2 and verify the attenuation of its virulence, as this pathogenicity island is required for the intracellular survival and replication of the bacteria necessary to cause systemic infection. Mutations in SPI2 result in a significant attenuation of Salmonella virulence and impair its ability to produce systemic infection due to its inability to survive within macrophages [28]). As shown in Figure 1, the survival of *S.* Typhimurium 14028s ∆SPI2 within murine RAW264.7 macrophages was significantly reduced at 24 and 48 h postinfection, compared to the wild-type *S.* Typhimurium 14028s strain.

### 3.2. Antigen Expression Detection via Western Blot Analysis

To detect the expression of the GnRXG/Q antigen in the recombinant strain *S.* Typhimurium 14028s ∆SPI2/pJEX, a Western blot analysis was performed. This technique relies on the recognition of proteins separated by polyacrylamide gel electrophoresis by specific antibodies after their transfer into nitrocellulose membranes. Figure 2 shows the correspondence of the GnRXG/Q antigen with the protein detected in the analysis. While antigen expression analysis for the pDNAX3 construct was not conducted in this preliminary study due to logistical constraints, we plan to include it in future work.

### 3.3. Antibody Levels Against GnRXG/Q

IgG anti-GnRXG/Q antibody titers were determined by an ELISA assay using the GnRXG/Q protein in 96-well ELISA plates. The antigen-antibody complex was detected using anti-mouse IgG conjugated to peroxidase (Jackson Immunoresearch). The groups orally vaccinated with strain *S.* Typhimurium 14028s ∆SPI2/pJEX, *S.* Typhimurium 14028s∆SPI2/pDNAX3, and the positive subcutaneous control group showed a significant increase in antibody levels on days 15, 30, 45, and 60 post immunization compared to the negative control group. Additionally, no significant differences were observed between the first three groups (Figure 3).

### 3.4. Measurement of Serum Testosterone Levels

Testosterone concentration in the serum of the study animals was measured using a commercial enzyme-linked immunosorbent assay (ELISA) kit for testosterone (IBL-AMERICA) on days 15, 30, and 45 post immunization. This solid-phase immunoassay is based on the principle of competitive binding, where an unknown amount of antigen in the samples competes with a fixed amount of enzyme-labeled antigen for the antibody binding sites. The wells were coated with a monoclonal antibody directed against a testosterone epitope, allowing the hormone in the serum samples to compete with a testosterone–peroxidase conjugate for binding to the immobilized antibody. After obtaining the absorbance of the known concentration standards and the serum samples, a standard curve was constructed with the mean absorbance of each standard and its concentration. The concentration for each sample was then determined using the standard curve, as presented in Figure 4.

Serum testosterone concentrations were evaluated at multiple time points post immunization to assess the impact of the treatments on hormonal levels. On day 15 (Figure 4A), a statistically significant reduction in testosterone levels was observed in both experimental groups compared to the negative control group, indicating the early effect of the immunization. However, the positive control group did not exhibit significant differences in testosterone levels relative to the negative control group at this time point. By day 30 (Figure 4B), only the positive control group demonstrated a significant decrease in serum testosterone when compared to the negative control group, while no significant differences were detected in the experimental groups, possibly reflecting a transient rebound effect. On day 45 (Figure 4C), both experimental groups, along with the positive control group, exhibited a marked and statistically significant reduction in testosterone levels relative to the negative control group, suggesting a sustained immunological response and the cumulative effect of the booster dose.

### 3.5. Histological Testicular Evaluation

The testes of the mice were extracted and fixed in 10% formalin, then sectioned at 5 µm thickness, stained with hematoxylin-eosin, and analyzed by optical microscopy. The images were obtained using a Nikon Eclipse E400 optical microscope (Nikon Instruments Inc., Melville, NY, USA) and Image-Pro Plus morphometric software (Bioimager Inc., ON, Canada). To demonstrate testicular tissue atrophy, the transverse tubular diameter of 10 seminiferous tubules and the number of epithelial cell layers in 10 seminiferous tubules were measured in each group (Table 1). Testicular tissue from the negative control group showed no signs of atrophy (Figure 5A). The experimental groups and the positive control group exhibited a significant reduction in the transverse tubular diameter of the seminiferous tubules, as well as a significant decrease in the number of seminiferous epithelial cell layers (Table 1 and Figure 5B–D).

## 4. Discussion

Previous studies have explored various strategies to generate an effective and safe immune response against Gonadotropin-Releasing Hormone (GnRH) in different animal species, achieving variable results. However, oral immunization had not been extensively investigated until now. The development of oral vaccine formulations offers several advantages over traditional injectable methods, including ease of administration, elimination of needles (enhancing operator safety), and relatively low production costs [23,39].

In this study, we utilized an attenuated *Salmonella Typhimurium* ΔSPI2 strain as an antigenic vector, which demonstrated safety due to its SPI2 mutation. This pathogenicity island is critical for bacterial survival within macrophages, while the intact SPI1 region preserves the strain’s ability to invade host cells and deliver heterologous genes. These characteristics make this vector highly suitable for the development of oral vaccines, simplifying administration and reducing handling challenges in animal management.

The attenuated *S.* Typhimurium strain was engineered to carry two plasmids encoding a modified GnRH gene (GnRXG/Q) with tandem repeats: one under prokaryotic expression (pJEX) and another under eukaryotic expression (pDNAX3). Both formulations induced a robust humoral immune response, as demonstrated by the significant increase in IgG anti-GnRXG/Q antibody titers in both experimental groups. These responses were comparable to those observed in the positive control group that received subcutaneous immunization, as shown in Figure 3. These findings confirm the efficacy of the *S.* Typhimurium vector in eliciting a potent immune response, aligning with prior evidence of its suitability as a delivery system for heterologous antigens demonstrated [23,40].

Notably, no adverse effects, including weight loss, were observed in the treated mice, underscoring the vaccine’s safety profile. Based on previously published data, the estimated colonization duration for the SPI2 mutant strain is approximately 15 days [41,42,43]. The deletion of SPI2 (or key genes located within this pathogenicity island) leads to a marked reduction in virulence in BALB/c mice, as well as in cattle, pigs, poultry, and humans [41,42,43,44]. Furthermore, vaccination with the SPI2 mutant has been shown to elicit slightly higher antibody production compared to wild-type strains, suggesting that this mutant strain may be a preferable option as a vector for delivering heterologous antigens, particularly when robust stimulation of the humoral immune response is desired [45]. The *S.* Typhimurium ΔSPI2 mutant used in this study is promising and offers a platform for adding extra safety features, such as including resistance-free inducible systems for regulating essential genes or auxotrophic-inducing mutations to ensure both clearing out of bacteria and efficient induction of immunity [46].

The testosterone analyses further highlight the vaccine’s immunological impact. On day 15 post immunization, both experimental groups showed a significant reduction in serum testosterone levels compared to the negative control group (Figure 4A). This indicates an early neutralization of GnRH by the induced antibodies. However, by day 30, testosterone levels rebounded in the experimental groups, while the positive control group maintained significantly lower levels compared to the negative control group (Figure 4B). This transient rebound may result from the physiological feedback loop, wherein reduced testosterone triggers hypothalamic GnRH secretion, leading to increased luteinizing hormone (LH) release and subsequent stimulation of Leydig cells to produce testosterone [47]. Following the booster dose administered on day 30, serum testosterone levels in the experimental groups decreased significantly again by day 45 (Figure 4C), demonstrating the booster’s ability to restore hormonal suppression. Future studies should explore whether administering the booster earlier than day 30 could sustain low testosterone levels over time.

Histological analyses provided further insights into the vaccine’s biological effects. Testicular tissue from the negative control group exhibited normal morphology, with no signs of atrophy (Figure 5A). In contrast, both experimental groups and the positive control group showed significant atrophy of the seminiferous epithelium, characterized by a marked reduction in the transverse tubular diameter and a decrease in the number of epithelial cell layers in the seminiferous tubules (Figure 5B–D, Table 1). These structural changes correlate with decreased testosterone levels and suggest impaired spermatogenesis. The reduction in tubular diameter and epithelial cell layers likely reflects a decline in Sertoli cell numbers and function, which are critical for maintaining the seminiferous epithelium and spermatogenesis [48,49]. While these findings strongly imply subfertility, fertility-specific studies are necessary to confirm the vaccine’s impact on reproductive capability.

Overall, this study demonstrates that oral immunization with *S.* Typhimurium ΔSPI2 carrying GnRXG/Q induces a robust immune response, effectively suppresses testosterone synthesis, and causes testicular atrophy in male mice. These results are consistent with those achieved via parenteral immunization [1,5,50,51]. Moreover, the use of *S.* Typhimurium ΔSPI2 as a vector represents a novel, safe, and practical approach for developing oral vaccines against GnRH.

Ongoing work is focused on enhancing the biosafety profile of this vaccine platform. For example, incorporating amino acid auxotrophy markers could eliminate the need for antibiotic resistance genes, reducing the risk of horizontal gene transfer and improving the vaccine’s environmental safety. Future investigations should evaluate the vaccine’s efficacy in female mice, its impact on fertility and fecundity in both sexes, and its application in domestic and companion animals.

These findings hold significant promise for veterinary medicine, offering a humane, cost-effective alternative to surgical castration and injectable immunocastration. By improving animal welfare and reducing economic barriers, this prototype of an oral vaccine has the potential to transform reproductive management practices in a variety of animal species.

## 5. Conclusions

Oral administration of *S.* Typhimurium 14028s ΔSPI2 transformed with two plasmids containing a tandem-repeated modified GnRH gene (*GnRXG/Q*), one for prokaryotic expression (*pJEX*) and one for eukaryotic expression (*pDNAX3*), proved to be effective in producing antibodies against *GnRXG/Q*, causing a decrease in serum testosterone levels and testicular tissue atrophy, as evidenced by a reduction in the transverse tubular diameter of the seminiferous tubules and the number of seminiferous epithelial cell layers in the testes of inoculated mice. The use of *Salmonella* ΔSPI2 is a feasible, safe, and simple system to produce an oral immunocastration formulation. This is the first approach using the strategy of oral immunization against GnRH. Further studies are needed to evaluate its effectiveness in female mice, the fertility and fecundity of male and female mice in vivo, and later, to assess its effectiveness in domestic and companion animals.

## Figures and Tables

**Figure 1 vaccines-12-01400-f001:**
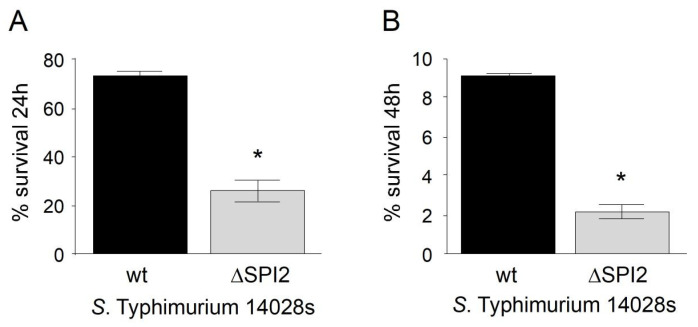
Survival percentage of *S.* Typhimurium 14028s and *S.* Typhimurium 14028s ∆SPI2 in murine RAW264.7 macrophages. (**A**) Survival at 24 h post internalization. (**B**) Survival at 48 h post internalization. An intracellular survival assay was performed, and intracellular bacteria were recovered at the indicated times. Survival percentages were calculated from the bacteria recovered at t_0_. The bars represent the standard deviation (SD). The assays were performed in triplicate. (*) Indicates a significant difference with *p* < 0.05 for the attenuated 14028s ∆SPI2 strain compared to the wild-type 14028s strain.

**Figure 2 vaccines-12-01400-f002:**
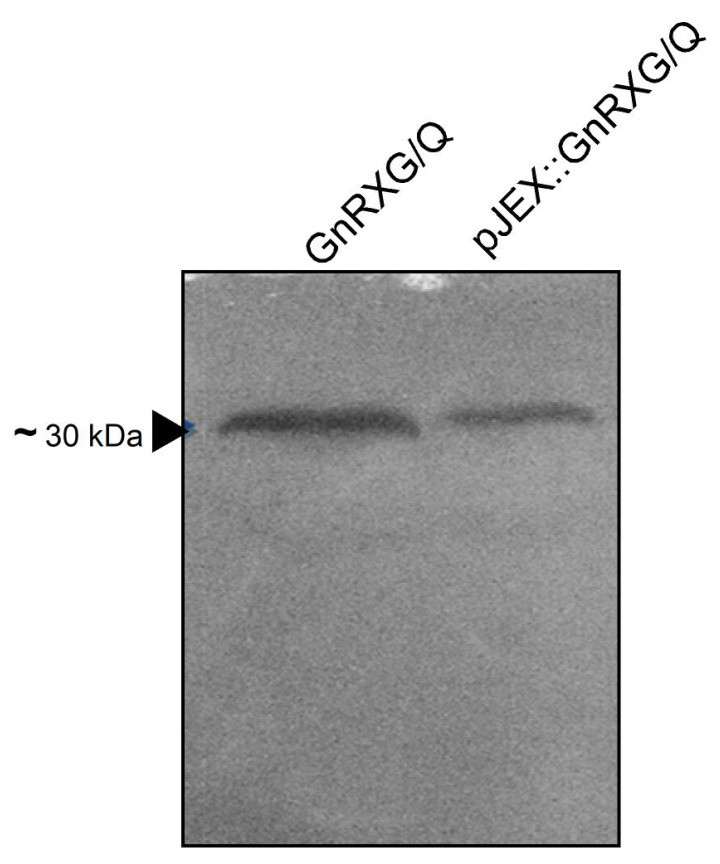
Detection of GnRXG/Q by Western blot. An isolated colony of the recombinant strain *S.* Typhimurium 14028s ∆SPI2/pJEX was cultured in LB broth and induced with IPTG. The purified GnRXG/Q antigen (~30 kDa) was used as a positive control. The extracts were analyzed by Western blot. The band was photographed under a transilluminator. Lane 1 corresponds to the positive control (GnRXG/Q purified protein), and Lane 2 corresponds to the protein expressed in the recombinant strain *S.* Typhimurium 14028s ∆SPI2/pJEX.

**Figure 3 vaccines-12-01400-f003:**
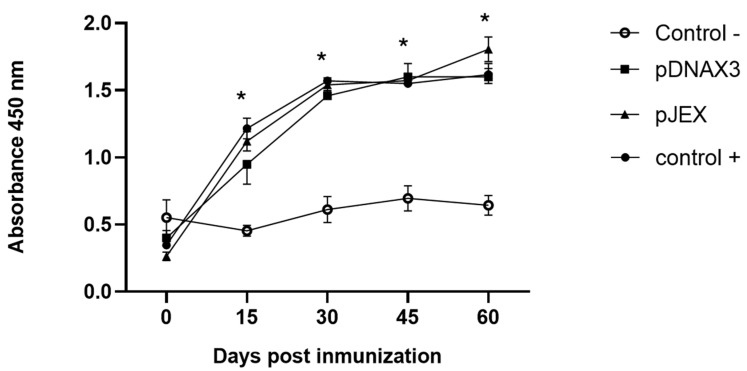
IgG anti-GnRXG/Q antibody levels expressed as arithmetic means of absorbance values measured at 450 nm on days 0, 15, 30, 45, and 60 post immunization. Male BALB/c mice (*n* = 3) were orally immunized on days 0 and 30 with 10^9^ CFU of *S.* Typhimurium 14028s ΔSPI2/pDNAX3::GnRXG/Q (pDNAX3) with 10^9^ CFU of *S.* Typhimurium 14028s ΔSPI2/pJEX::GnRXG/Q (pJEX), subcutaneously with 100 µg of GnRXG/Q purified by affinity chromatography using aluminum as an adjuvant (Control +), and orally with 200 µL of PBS (Control −). IgG anti-GnRXG/Q levels were measured by ELISA on days 0, 15, 30, 45, and 60 post immunization. (*) Indicates a significant difference with *p* < 0.05 for any of the treated groups compared with the negative control.

**Figure 4 vaccines-12-01400-f004:**
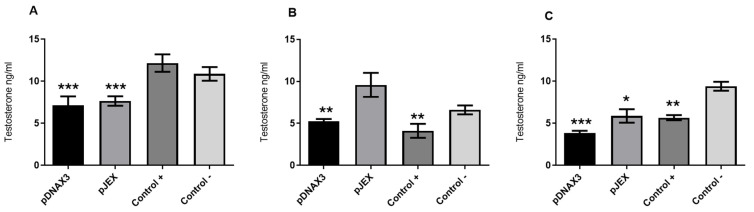
Serum testosterone concentration of the study animals, based on a standard curve. (**A**) Day 15 post immunization. (**B**) Day 30 post immunization. (**C**) Day 45 post immunization. Male BALB/c mice (*n* = 3) were orally immunized on days 0 and 30 with 10^9^ CFU of *S.* Typhimurium 14028s ΔSPI2/pDNAX3::GnRXG/Q (pDNAX3) with 10^9^ CFU of *S.* Typhimurium 14028s ΔSPI2/pJEX::GnRXG/Q (pJEX), subcutaneously with 100 µg of GnRXG/Q purified by affinity chromatography using aluminum as an adjuvant (Control +), and orally with 200 µL of PBS (Control −). Serum testosterone levels were measured by commercial ELISA (IBL-AMERICA kit) on days 15, 30, and 45 post immunization. (***) Indicates a difference of *p* < 0.001 compared to the negative control. (**) Indicates a difference of *p* < 0.01 compared to the negative control. (*) Indicates a difference of *p* < 0.05 compared to the negative control.

**Figure 5 vaccines-12-01400-f005:**
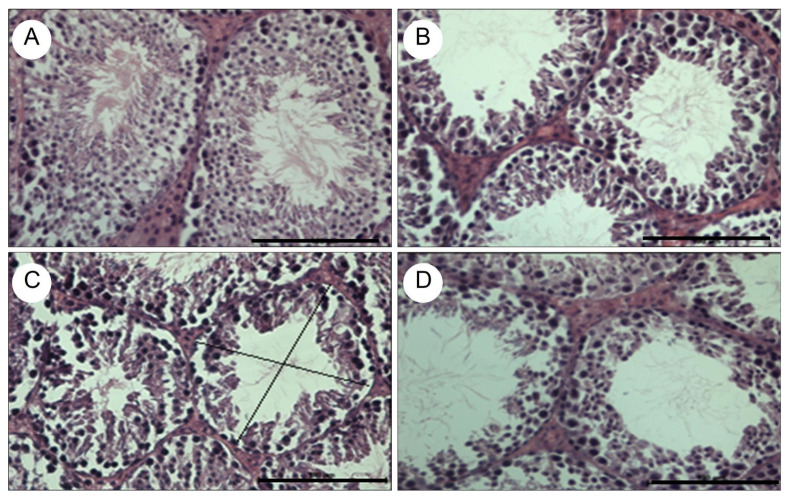
Histological sections of seminiferous tubules, stained with hematoxylin-eosin, observed at 40×. The images correspond to the negative control group immunized with 200 μL of oral PBS (**A**), the positive control group immunized subcutaneously with 100 μg of GnRXG/Q purified by aluminum affinity chromatography as an adjuvant (**B**), the experimental group orally immunized with strain *S.* Typhimurium 14028s ∆SPI2/pDNAX3, (**C**) and the experimental group immunized orally with strain *S.* Typhimurium 14028s ∆SPI2/pJEX (**D**). The black lines show the transverse tubular diameter, represented by the means of two diametrically opposed measurements. The black solid bar at the lower right corner of each figure is 100 µm long.

**Table 1 vaccines-12-01400-t001:** Transverse tubular diameter and number of seminiferous epithelial cell layers in the testes of the study mice.

Group	Transverse Tubulardiameter (µm) ± SEM	No. of Seminiferous Epithelial Cell Layers ± SEM
Control −	206.485 ± 21.69	6.3 ± 0.57
Control +	142.908 ± 18.90 ***	4.26 ± 0.63 ***
pDNAX3	128.625 ± 21.99 ***	3.76 ± 0.56 ***
pJEX	137.85 ± 13.27 ***	4.7 ± 0.59 ***

Histological evaluation of seminiferous tubules and epithelial cells from the testes of the study mice. Values are expressed as mean ± SEM. The transverse tubular diameter corresponds to the average of two diametrically opposed measurements. (***) Indicates a significant difference with *p* < 0.001 compared to the negative control group.

## Data Availability

The datasets used and/or analysed during the current study are available from the corresponding author on reasonable request.

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
