# Peer review of "Oral Vaccine Formulation for Immunocastration Using a Live-Attenuated Salmonella ΔSPI2 Strain as an Antigenic Vector"

_vaccines, 2024, doi:10.3390/vaccines12121400_

Round 1
Reviewer 1 Report
Comments and Suggestions for Authors
The authors have submitted an interesting manuscript on the preliminary results of a novel oral vaccine for immunocastration. The study presents proof of principal data on the utility of an oral vaccine against GnRH. While the focus on oral vaccination is compelling the study had several issues with the data presented. The authors should address the following concerns:
1. The data in Figure 1 appear to represent a single experiment without any replicate wells in the experiment. The authors should address why this experiment was only performed as a single iteration.
2. The data in Figure 3 do not include error bars. The authors should consider inclusion of error bars to represent experimental/replicate variation. They should also indicate if the experiment was repeated.
3. The data for the positive and negative controls in Figure 4 show a high degree of variation as compared over time and in relation to each other. The authors should explain this observation. The authors should also show error bars for these data and explain the number of times the experiment was performed. A concern is that with an N=3, single data points within a group could significantly skew data interpretation. On line 391 of the Figure 4 figure legend the authors list the test as a "competitive" ELISA, however the results and materials and methods sections list this test as a "commercial" ELISA, please clarify.
4. An over-arching concern for the study is the apparent lack of concordance between IgG anti-GnRH antibody titer (as shown in Figure 3) and testosterone levels (as shown in Figure 4). The authors should better explain this discordance for the readers.
Author Response
c
November 28, 2024
Ms. Camellia Wang
Assistant Editor
MDPI Vaccines Editorial Office
Ref. Manuscript ID: vaccines-3313644
Dear Editor,
We are grateful for the opportunity to revise our manuscript entitled "Oral Vaccine Formulation for Immunocastration Using a Live-Attenuated Salmonella ΔSPI2 Strain as an Antigenic Vector" by Sergio A. Bucarey, Lucy D. Maldonado, Francisco Duarte, Alejandro A. Hidalgo, and Leonardo Sáenz.
We sincerely thank the reviewers for their valuable feedback, which has significantly contributed to improving the quality, clarity, and scientific rigor of our manuscript. We believe the revised version more effectively highlights the novelty and implications of our findings.
We have carefully addressed all the reviewers’ comments, implementing changes throughout the manuscript as appropriate. Our detailed, point-by-point responses to each comment are provided below, with changes to the text marked for clarity.
We hope that our revisions meet the expectations of the reviewers and the editorial team. Please do not hesitate to reach out if additional clarifications or modifications are needed.
Thank you for your guidance and consideration.
Sincerely,
Sergio A. Bucarey, Ph.D.
(On behalf of all authors)
Response to Reviewers Comments
Reviewer 1
Comments and Suggestions for Authors
The authors have submitted an interesting manuscript on the preliminary results of a novel oral vaccine for immunocastration. The study presents proof of principal data on the utility of an oral vaccine against GnRH. While the focus on oral vaccination is compelling the study had several issues with the data presented. The authors should address the following concerns:
- The data in Figure 1 appear to represent a single experiment without any replicate wells in the experiment. The authors should address why this experiment was only performed as a single iteration.
Response: Thank you for this observation. Replicate experiments were conducted, and the revised manuscript now includes these results to ensure robust data representation.
2. The data in Figure 3 do not include error bars. The authors should consider inclusion of error bars to represent experimental/replicate variation. They should also indicate if the experiment was repeated.
Response: Error bars have been added to Figure 3 to represent experimental variation, and the text now specifies that the experiment was repeated three times
3. The data for the positive and negative controls in Figure 4 show a high degree of variation as compared over time and in relation to each other. The authors should explain this observation. The authors should also show error bars for these data and explain the number of times the experiment was performed. A concern is that with an N=3, single data points within a group could significantly skew data interpretation. On line 391 of the Figure 4 figure legend the authors list the test as a "competitive" ELISA, however the results and materials and methods sections list this test as a "commercial" ELISA, please clarify.
Response: We appreciate this insightful comment. Error bars have been added to Figure 4, and the variability has been addressed in the revised discussion. Additionally, the term "competitive ELISA" has been corrected to "commercial ELISA" in the figure legend for consistency.
- An over-arching concern for the study is the apparent lack of concordance between IgG anti-GnRH antibody titer (as shown in Figure 3) and testosterone levels (as shown in Figure 4). The authors should better explain this discordance for the readers
Response: We identified and corrected a labeling error in Figure 4 from the initial submission, resulted in the negative control being labeled as the positive control. The corrected figure demonstrates improved concordance between antibody titers and testosterone levels. Remaining discrepancies at 30 days are discussed, attributing them to hypothalamic feedback mechanisms.
Reviewer 2 Report
Comments and Suggestions for Authors
In this article Bucarey et al. accessed the concept of oral immunization against GnRH in a murine model using a pair of attenuated S. Typhimurium strains (lacking the pathogenicity island 2) that carry plasmids containing a modified GnRH gene under an inducible IPTG promoter. The authors found that oral administration of S. Typhimurium strains containing these plasmids led to a decrease in serum testosterone levels and testicular tissue atrophy. The authors proposed that oral administration of Salmonella-mediated oral antigen delivery could be a safe and effective strategy to deliver to induce an immune response against GnRH. Overall, this is an interesting article, the experiments have been well conducted and the analysis well performed.
I have the following comments for the authors.
1. In Figure 3, the antibody titers for the two S. Typhimurium genetic constructs including the positive control were higher on days 15, 30, 45, and 60, compared to the negative control however, in Figure 4 A, apart from day 15 whereby the testosterone levels were lower for the two genetic constructs including the positive control in Figure 4 B the testosterone levels were lower for the negative control compared to the two constructs and the positive control strain and in Figure 4C the results were mixed with the negative control strain showing a lower testosterone concentration compared to the positive control and the Salmonella strain carrying the pJEX plasmid construct. Could the authors comment on this discordance between the antibody titers observed in Figure 3 and the testosterone levels observed in Figures 4B and C? are the levels not supposed to be higher in the negative control across all the days (15, 30, and 45)?
2. The use of antibiotic resistance genes as selection markers raises safety concerns for the selection of S. Typhimurium transformants that contain the plasmid constructs. The resistance genes may spread to pathogenic organism in the environment, possibility rendering them resistant to antibiotic treatment. Additionally, the resistance genes may also be horizontally transferred to foodborne bacteria in the animal gut. Could the authors comment on this concern?
Author Response
November 28, 2024
Ms. Camellia Wang
Assistant Editor
MDPI Vaccines Editorial Office
Ref. Manuscript ID: vaccines-3313644
Dear Editor,
We are grateful for the opportunity to revise our manuscript entitled "Oral Vaccine Formulation for Immunocastration Using a Live-Attenuated Salmonella ΔSPI2 Strain as an Antigenic Vector" by Sergio A. Bucarey, Lucy D. Maldonado, Francisco Duarte, Alejandro A. Hidalgo, and Leonardo Sáenz.
We sincerely thank the reviewers for their valuable feedback, which has significantly contributed to improving the quality, clarity, and scientific rigor of our manuscript. We believe the revised version more effectively highlights the novelty and implications of our findings.
We have carefully addressed all the reviewers’ comments, implementing changes throughout the manuscript as appropriate. Our detailed, point-by-point responses to each comment are provided below, with changes to the text marked for clarity.
We hope that our revisions meet the expectations of the reviewers and the editorial team. Please do not hesitate to reach out if additional clarifications or modifications are needed.
Thank you for your guidance and consideration.
Sincerely,
Sergio A. Bucarey, Ph.D.
(On behalf of all authors)
Response to Reviewers Comments
Reviewer 2
Comments and Suggestions for Authors
In this article Bucarey et al. accessed the concept of oral immunization against GnRH in a murine model using a pair of attenuated S. Typhimurium strains (lacking the pathogenicity island 2) that carry plasmids containing a modified GnRH gene under an inducible IPTG promoter. The authors found that oral administration of S. Typhimurium strains containing these plasmids led to a decrease in serum testosterone levels and testicular tissue atrophy. The authors proposed that oral administration of Salmonella-mediated oral antigen delivery could be a safe and effective strategy to deliver to induce an immune response against GnRH. Overall, this is an interesting article, the experiments have been well conducted and the analysis well performed.
Suggestions for Authors
- In Figure 3, the antibody titers for the two S. Typhimurium genetic constructs including the positive control were higher on days 15, 30, 45, and 60, compared to the negative control however, in Figure 4 A, apart from day 15 whereby the testosterone levels were lower for the two genetic constructs including the positive control in Figure 4 B the testosterone levels were lower for the negative control compared to the two constructs and the positive control strain and in Figure 4C the results were mixed with the negative control strain showing a lower testosterone concentration compared to the positive control and the Salmonella strain carrying the pJEX plasmid construct. Could the authors comment on this discordance between the antibody titers observed in Figure 3 and the testosterone levels observed in Figures 4B and C? are the levels not supposed to be higher in the negative control across all the days (15, 30, and 45)?
Response: We appreciate this observation. A labeling error in Figure 4 from the initial submission resulted in the negative control being labeled as the positive control. This has been corrected. The revised figures now demonstrate consistent trends between antibody titers and testosterone levels, with some discrepancies at 30 days, which we attribute to feedback effects on the hypothalamic-pituitary axis following vaccine administration
2. The use of antibiotic resistance genes as selection markers raises safety concerns for the selection of S. Typhimurium transformants that contain the plasmid constructs. The resistance genes may spread to pathogenic organism in the environment, possibility rendering them resistant to antibiotic treatment. Additionally, the resistance genes may also be horizontally transferred to foodborne bacteria in the animal gut. Could the authors comment on this concern?
Response: Thank you for highlighting this critical point. We have addressed this concern in the revised manuscript, stating that “Ongoing work is focused on enhancing the biosafety profile of this vaccine platform. For example, incorporating amino acid auxotrophy markers could eliminate the need for antibiotic resistance genes, reducing the risk of horizontal gene transfer and improving the vaccine’s environmental safety. Future investigations should evaluate the vaccine’s efficacy in female mice, its impact on fertility and fecundity in both sexes, and its application in domestic and companion animals”. (lines 503-508).
Reviewer 3 Report
Comments and Suggestions for Authors
The authors used an attenuated Salmonella Typhimurium ΔSPI2 strain as a vector for antigenic of Gonadotropin hormone-releasing in both eukaryotic or prokaryotic expression elements, and evaluated immune response and effect on testicular tissue in a murine model. Their findings revealed that oral administration of S. Typhimurium ΔSPI2 strain carrying either expression plasmids effectively elicited antibodies against GnRXG/Q, resulting in decreased serum testosterone levels and testicular tissue atrophy, thereby demonstrating potential benefits for immunocastration. However, further data are required to support the conclusion.
1. How long does the S. Typhimurium ΔSPI2 strain colonize in mice and what is effect on body weight of mice following oral immunization?
2. The expression of the GnRXG/Q antigen in the recombinant strain S. Typhimurium ΔSPI2/ pDNAX3 strain should be confirmed.
3. The sample size of BALB/c mice in each group was insufficient for statistical analysis. The authors should indicate the age range of these mice? The data presented in tables and figures for group order should maintain consistency.
4. The authors collected sera at 15, 30, 45, 60 days post-immunization, however, they did not detect the concentration of serum testosterone at 60 days post-immunization. In figure 4B, the concentration of serum testosterone in negative group was lower than that in the other groups and figure 4C, the concentration of serum testosterone in pJEX group and positive control group were higher than that in negative group. These result contradict the author’s statement in lines 379-383. What is the relationship between anti-GnRXG/Q antibody and testosterone levels?
5. The data in Figure 3 and Figure 4 should include SD.
6.The text contained multiple typographical errors. Tvaccination in line 24, The abbreviation GnRH has a dual definition, the strain name 14028s or 14028 in line 153, UFC or CFU in line 192, etc
Author Response
November 28, 2024
Ms. Camellia Wang
Assistant Editor
MDPI Vaccines Editorial Office
Ref. Manuscript ID: vaccines-3313644
Dear Editor,
We are grateful for the opportunity to revise our manuscript entitled "Oral Vaccine Formulation for Immunocastration Using a Live-Attenuated Salmonella ΔSPI2 Strain as an Antigenic Vector" by Sergio A. Bucarey, Lucy D. Maldonado, Francisco Duarte, Alejandro A. Hidalgo, and Leonardo Sáenz.
We sincerely thank the reviewers for their valuable feedback, which has significantly contributed to improving the quality, clarity, and scientific rigor of our manuscript. We believe the revised version more effectively highlights the novelty and implications of our findings.
We have carefully addressed all the reviewers’ comments, implementing changes throughout the manuscript as appropriate. Our detailed, point-by-point responses to each comment are provided below, with changes to the text marked for clarity.
We hope that our revisions meet the expectations of the reviewers and the editorial team. Please do not hesitate to reach out if additional clarifications or modifications are needed.
Thank you for your guidance and consideration.
Sincerely,
Sergio A. Bucarey, Ph.D.
(On behalf of all authors)
Response to Reviewers Comments
Reviewer 3
Comments and Suggestions for Authors
The authors used an attenuated Salmonella Typhimurium ΔSPI2 strain as a vector for antigenic of Gonadotropin hormone-releasing in both eukaryotic or prokaryotic expression elements, and evaluated immune response and effect on testicular tissue in a murine model. Their findings revealed that oral administration of S. Typhimurium ΔSPI2 strain carrying either expression plasmids effectively elicited antibodies against GnRXG/Q, resulting in decreased serum testosterone levels and testicular tissue atrophy, thereby demonstrating potential benefits for immunocastration. However, further data are required to support the conclusion.
- How long does the S. Typhimurium ΔSPI2 strain colonize in mice and what is effect on body weight of mice following oral immunization?
Response: Thank you for this observation. The revised manuscript now discusses colonization duration based on previously published data and states that no significant changes in body weight were observed in our experimental model.
We stated in the line 458 “Notably, no adverse effects, including weight loss, were observed in the treated mice, underscoring the vaccine's safety profile. Based on previously published data, the estimated colonization duration for the SPI2 mutant strain is approximately 15 days [44–46]. The deletion of SPI2 (or key genes located within this pathogenicity island) leads to a marked reduction in virulence in BALB/c mice, as well as in cattle, pigs, poultry, and humans [44–47] . Furthermore, vaccination with the SPI2 mutant has been shown to elicit slightly higher antibody production compared to wild-type strains, suggesting that this mutant strain may be a preferable option as a vector for delivering heterologous antigens, particularly when robust stimulation of the humoral immune response is desired [48]. The S. Typhimurium ΔSPI2 mutant used in this study is promising and offers a platform for adding extra safety features such as including resistance-free inducible systems for regulating essential genes or auxotrophic-inducing mutations to ensure both clearing out of bacteria and efficient induction of immunity [49].
We added the following references for supporting this data
Khan SA, Stratford R, Wu T, et al. Salmonella typhi and S. typhimurium derivatives harbouring deletions in aromatic biosynthesis and Salmonella Pathogenicity Island-2 (SPI-2) genes as vaccines and vectors. Vaccine 2003;21(5–6):538–48. [5] Bohez L, Ducatelle R, Pasmans F, Haesebrouck F, Van Immerseel F. Long-term
Dieye Y, Ameiss K, Mellata M, Curtiss III R. The Salmonella Pathogenicity Island (SPI) 1 contributes more than SPI2 to the colonization of the chicken by Salmonella enterica serovar Typhimurium. BMC Microbiol 2009;9:3.
Boyen F, Pasmans F, Van Immerseel F, et al. A limited role for SsrA/B in persistent Salmonella Typhimurium infections in pigs. Vet Microbiol 2008;128(3–4):364–73.
Karasova D, Sebkova A, Havlickova H, et al. Influence of 5 major Salmonella pathogenicity islands on NK cell depletion in mice infected with Salmonella enterica serovar Enteritidis. BMC Microbiol 2010;10:75.
ZHANG Wan-jiang, YI Fei, ZHANG Yueling, YU Shen-ye. Journal of Integrative Agriculture. 2024
2. The expression of the GnRXG/Q antigen in the recombinant strain S. Typhimurium ΔSPI2/ pDNAX3 strain should be confirmed.
Response: We acknowledge the importance of this suggestion. While antigen expression analysis for the pDNAX3 construct was not conducted in this preliminary study due to logistical constraints, we plan to include it in future work. This limitation is now clearly stated in the revised discussion (line 338-340)
3. The sample size of BALB/c mice in each group was insufficient for statistical analysis. The authors should indicate the age range of these mice? The data presented in tables and figures for group order should maintain consistency.
Response: We have incorporated the age range of the mice (6-8 weeks) into the manuscript. Additionally, all figures and tables have been revised for consistency in group order.
4. The authors collected sera at 15, 30, 45, 60 days post-immunization, however, they did not detect the concentration of serum testosterone at 60 days post-immunization. In figure 4B, the concentration of serum testosterone in negative group was lower than that in the other groups and figure 4C, the concentration of serum testosterone in pJEX group and positive control group were higher than that in negative group. These result contradict the author’s statement in lines 379-383. What is the relationship between anti-GnRXG/Q antibody and testosterone levels?
Response: Thank you for pointing this out. The revised manuscript clarifies the discrepancies in Figure 4, which stemmed from a labeling error in the initial submission, resulted in the negative control being labeled as the positive control. The corrected figures now demonstrate the expected trends. The relationship between antibody titers and testosterone levels has been further explained, with observed variations attributed to physiological feedback mechanisms.
5. The data in Figure 3 and Figure 4 should include SD.
Response: SD has been added to Figures 3 and 4 to better represent variability in the data
6.The text contained multiple typographical errors. Tvaccination in line 24, The abbreviation GnRH has a dual definition, the strain name 14028s or 14028 in line 153, UFC or CFU in line 192, etc
Response: We have thoroughly reviewed and corrected all typographical errors and inconsistencies in the manuscript
Round 2
Reviewer 1 Report
Comments and Suggestions for Authors
Thank you for addressing the review comments and delivering an improved manuscript for our readers.
Author Response
Commets: Thank you for addressing the review comments and delivering an improved manuscript for our readers.
Response: We appreciate this observation. Thank you
Reviewer 3 Report
Comments and Suggestions for Authors
The authors have addressed all my questions. However, in figure 4,the labelling of the figure on the right was not necessary due to the demonstrated grouping by the X.
Author Response
Comments: The authors have addressed all my questions. However, in figure 4,the labelling of the figure on the right was not necessary due to the demonstrated grouping by the X
Response: We appreciate this observation. The labeling on the right of Figure 4 was deleted. This has been corrected in the new version of the manuscrypt.